# Development of Nutrient Solution Compositions for Paprika Cultivation in a Closed Coir Substrate Hydroponic System in Republic of Korea's Winter Cropping Season

Kyung-Hwan Yeo [1,*], Gyeong-Lee Choi [2], Jae-Han Lee [1], Kyung-Sub Park [3] and Ki-Young Choi [4,*]

1. Vegetable Research Division, National Institute of Horticultural and Herbal Science, Rural Development Administration, Wanju 55365, Republic of Korea
2. Protected Horticulture Research Institute, National Institute of Horticultural and Herbal Science, Rural Development Administration, Haman 52054, Republic of Korea
3. Department of Horticultural Science, Mokpo National University, Muan 58554, Republic of Korea
4. Division of Future Agriculture Convergence, College of Agriculture and Life Sciences, Kangwon National University, Chuncheon 24341, Republic of Korea
* Correspondence: khyeo@korea.kr (K.-H.Y.); choiky@kangwon.ac.kr (K.-Y.C.); Tel.: +82-63-238-6663 (K.-H.Y.); +82-33-250-7771 (K.-Y.C.)

**Abstract:** Other than in highland areas, the majority of paprika growers in Republic of Korea transplant during late summer and cultivate when temperatures are relatively low. Additionally, they typically apply European nutrient solutions and hydroponic technologies with different climates and cultivation periods. Therefore, this study was conducted to determine the optimal nutrient solution composition based on growth stages suitable for paprika (*Capsicum annum* L.) cultivation in closed hydroponic systems with a coir substrate during the winter cropping season. The nutrient solutions were supplied to all the paprika plants at pH 5.8–6.0 and electrical conductivity (EC) 2.0–3.5 dS·m$^{-1}$ according to the target EC setting for each growth stage. The mixing ratio of the reused nutrient solution (drainage) to the fresh solution was maintained within 30% on an EC basis (0.6–1.1 dS·m$^{-1}$) at each irrigation event. Paprika plants 'Cupra' were grown in three different nutrient solutions using an automated fertilizer dosing system based on integrated solar radiation (ISR). Nutrient absorption patterns according to growth stages were investigated for each nutrient solution. To reduce the nutritional imbalances of major nutrients in the root zone (RZ, substrate), the nutrient solution 'NIHHS (National Institute of Horticultural and Herbal Sciences)' was adjusted considering the nutrition absorption patterns according to growth stages, ion balances between anions and cations, and differences between the ion concentrations in the RZ and with irrigation. NIHHS-Coir1 and NIHHS-Coir2 solutions with 5–10% and 10–15% readjustments of macronutrient concentrations in the NIHHS, respectively, were used to evaluate the suitabilities for cultivation and productivity, whilst the IKC (Informatie en Kennis Centrum) nutrient solution was used as the control. We investigated the influence of these solutions on growth, yield, and photosynthetic responses compared to the conventional nutrient solution. Throughout the entire cultivation period, the newly formulated NIHHS-Coir1 solution had a marketable fruit ratio (%) of 2.4–3.5% higher and a marketable yield (kg/m$^2$) of 5.3–8.7% higher than those of the conventional IKC solution.

**Keywords:** recirculating nutrient solution; coconut coir; irrigation; root zone



## 1. Introduction

The closed hydroponic system prevents soil salt accumulation and groundwater contamination, protects water resources, and reduces fertilizer consumption by reusing the circulated nutrient solution [1]. A previous study found that the water use for closed hydroponics in paprika plants was 7229 tons/ha/year, which is 29% less than that for noncirculated hydroponics, and that the annual fertilizer uses of N, P, K, Ca, Mg, and

S were reduced by 47, 39, 44, 55, and 56%, respectively [2]. To manage nutrients in this system, it is best to directly control the concentration of individual ions in the drainage solution [3,4], but this has not been practiced in Korean closed hydroponic farms. Instead of controlling individual ions, the electrical conductivity (EC) of the drainage solution is measured and diluted to the target EC level.

In a closed hydroponic system that controls nutrients based on the EC of the drainage solution, the automatic mixing of fresh and reused nutrient solutions (the drainage) at a fixed ratio has been widely employed commercially [5]. However, continuous drainage reuse in a recirculating hydroponic system results in considerable imbalances in nutrient ion ratios and the accumulation of bivalent ions such as Ca, Mg, and $SO_4$ [4,6]. Na and Cl are also representative ions which accumulate in drainage during the paprika cultivation period in the winter cropping season [7]. Under EC-based nutrient control in a closed-loop hydroponic system for paprika, the ionic contribution to EC of recirculated nutrient solution was found to be highest in K and $NO_3$-N, followed by Ca, $SO_4$-S, and Mg [8].

Jang et al. [9] reported that the growth and relative yields (%) of summer paprika in an EC-based closed hydroponic system decreased with increasing drainage mixing ratio in the early growth stages compared to the control supplied by a nutrient solution under no drainage reuse; however, after the midterm growth stage, the reuse of the drainage solution did not negatively affect plant growth and fruit quality. In a recent study, there were no significant differences found in the growth or fruit quality of paprika according to the type of substrate in closed rockwool or a coir substrate hydroponic system, where 30–40% of the irrigation water was drained whilst the EC of the reused drain was set at 1.5 dS·m$^{-1}$ [7].

Paprika was introduced in Republic of Korea in the early 1990s following the advanced use of glass greenhouses. The cultivation area and production yield increase every year, and were 733 ha and 81,841 tons as of 2020, respectively, with 44% of the total production exported [10]. Paprika provides a total income of $32.9 per unit area (m$^2$), making it one of the high-income greenhouse crops in Korea, whilst also having the highest ratio of hydroponics to total cultivation area [10,11]. Korean paprika farms currently use European nutrient management technology, which was developed for use in areas with a different climate (Marine West Coast) and cropping season. Nutrient solution formulations that are suitable for paprika cultivation in the climate of Northeast Asia have not been developed.

When paprika is cultivated on flat regions of Korea during summer, the insides of greenhouses become very humid, with the temperature rising above 35 °C, making it very difficult to manage the environment; as a result, fruit-setting is not carried out normally, and the incidences of nonmarketable fruits such as BER and sunburn are also high [12]. In order to effectively improve the greenhouse environment during summer, heat pumps, fogs, fans, and root zone cooling can all be used, but the initial installation costs and energy consumptions for these are high [13,14]. Therefore, many paprika growers, with the exception of those in the highland areas in Republic of Korea, transplant paprika in August or September and cultivate it until late June or early July of the following year to circumvent high temperatures. Recently, coconut coir has been extensively used as an eco-friendly organic medium that is easy to recycle and dispose of and that can replace peat moss in a single or mixed form with other substrates.

There has been rapid increase in the use of coir media in greenhouses in Republic of Korea since the early 2000s because the raw materials are easily available, affordable, and eco-friendly. When the coir substrate was introduced in Korea, its quality was not uniform, and it was difficult to control moisture contents according to the growth stage compared to the rockwool [15]. However, the addition of crushed chips to the coir resulted in increased porosity, improved uniformity as drainage was facilitated, and also made it easier to control the moisture content in the root zone [7,16]. This technique is currently most commonly used in paprika hydroponic cultivation [10]. However, standard nutrient solution formulations for closed hydroponics in paprika coir cultivation suitable for the climate in Northeast Asia have not been developed, and the productivity of greenhouse paprika in Republic of Korea is lower than that in Europe [10].

Peppers are fruit vegetable crops that develop both vegetative and generative growth simultaneously, with maintenance of optimal growth being closely related to maximizing yields [17]. Methods of controlling physiological plant balance include climate and irrigation control, as well as crop management; of these, irrigation is the most powerful means for controlling productivity and fruit quality through the supply of moisture and nutrients [18]. In the closed hydroponic system that continuously reuses nutrient solutions, the concentration of inorganic ions in the recirculating solution differs from that of the initially supplied irrigation solution. In addition, inorganic ions may be adsorbed to or discharged from the substrate according to the characteristics of the medium during hydroponic cultivation. The coir substrate has a relatively high cation exchange capacity (CEC), therefore it easily adsorb nutrients, and it also has high contents of Na, Cl, K, and P, whilst some of these may be discharged out of the medium. However, research into the changes of inorganic ions in the recirculated nutrient solution in the closed hydroponic system is still insufficient [19,20]. In order to maintain an optimal root environment, it is necessary to identify the nutrient absorption patterns of paprika by monitoring the concentrations of inorganic ions in the root zone in accordance with the growth stage, whilst also supplying a nutrient solution suitable for the needs of crops.

Therefore, this study was conducted to determine the optimal nutrient solution composition for paprika cultivation in closed hydroponic systems with a coir substrate by analyzing the nutrient absorption pattern according to growth stages during the winter cropping season.

## 2. Materials and Methods

### 2.1. Experimental Site and Conditions

This study was conducted in a Venlo-type glasshouse (Figure 1) located at the Protected Horticulture Research Institute, National Institute of Horticultural and Herbal Sciences (NIHHS) of Republic of Korea (latitude 35°24′N and longitude 128°42′E), from August 2014 to July 2017. The study was conducted using three experiments, with three cultivation cycles, over three years. The first experiment (Exp. 1) began on 26 August 2014 and ended on 20 July 2015 for the first cycle, whilst the second cycle for the second experiment (Exp. 2) began on 24 August 2015 and ended on 21 July 2016. Furthermore, the third experiment (Exp. 3) began on 18 August 2016 and ended on 20 July 2017 for the third cycle. In terms of temperature management in the greenhouse, after the transplant, the daytime and nighttime temperatures were set at 23–27 °C and 21 °C, respectively, using ventilation from the roof windows and heating with hot water circulating at 40–50 °C. During the winter season, the nighttime target temperature was set at 18 °C, with the temperature in the greenhouse being raised by 1 °C per hour from 4 a.m. and subsequently maintained at 21 °C until 9 a.m. In order to control solar radiation entering the greenhouse, 80% of the radiation was shielded using a shading and energy-saving screen (PH 77(B), Phormium Lokeren, Belgium), when the internal light intensity was 500 W·m$^{-2}$ or more, and 80% of the radiation was shielded with a sun shading screen (PH 55O, Phormium, Lokeren, Belgium) when the light intensity was 800 W·m$^{-2}$ or more.

### 2.2. Plant Materials

Paprika seeds ('Cupra', Enza Zaden, The Netherlands) (Figure 1) were sown in rockwool plug trays (CultiOne 240, Cultilene, The Netherlands) on 26 August 2014 for Exp. 1, 24 August 2015 for Exp. 2, and 18 August 2016 for Exp. 3, and the seedlings with 2–3 leaves were planted on the rockwool cubes (10 × 10 × 6.5 cm, UR Rockwool, Republic of Korea) sufficiently saturated with the standard nutrient solution for paprika rockwool hydroponics formulated by the Research Station for Floriculture and Glasshouse Vegetables (PBG; Proefstation voor Bloemisterij en Glasgroente) at Naaldwijk in The Netherlands. The paprika seedlings were transplanted onto desalinized coconut coir slabs (90 × 15 × 7.5 cm, chip:dust (*v:v*) = 5:5, Daeyoung GS, Republic of Korea) at a density of 2.87 plants·m$^{-2}$ on 23 September 2014 for Exp. 1, 24 September 2015 for Exp. 2, and 20 September 2016 for Exp.

3. The coir slab used in this study had a container capacity (%) of 50.5, as well as air space (%) of 1.2, total porosity (%) of 56.0, and bulk density (g·m$^{-3}$) of 0.066.

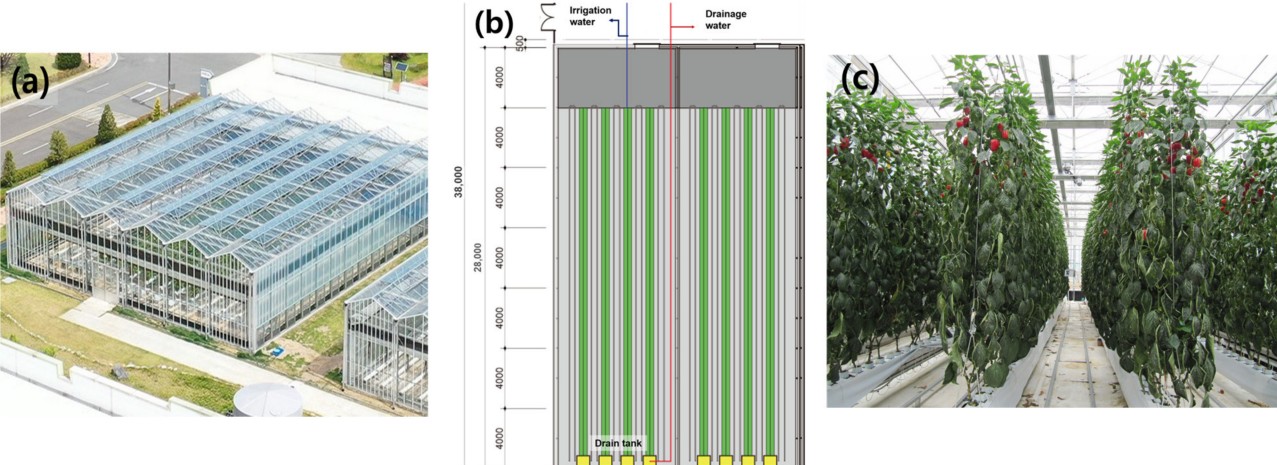

**Figure 1.** The Venlo-type glasshouse (**a**), its internal details (**b**), and paprika (*Capsicum annum* L. var. Cupra) plants (**c**) used in the experiment.

### 2.3. Closed Hydroponics System Setup

Paprika (*Capsicum annum* L. var. Cupra) plants were grown in three types of recirculating nutrient solutions for each experiment using an automated fertilizer dosing system (Nutrijet, Priva, The Netherlands) and an irrigation system (SH-1700, Shinhan A-TEC, Republic of Korea) based on integrated solar radiation (ISR) (Figure 2). The set ISR value was adjusted within the range of 60–250 J·cm$^{-2}$ depending on the plant growth stage and drainage ratio (%). Drip irrigation was initiated when the ISR inside the greenhouse reached the set value. The drainage ratio (%, drainage amount/irrigation amount × 100) was controlled at 10 to 40% according to the plant growth stages and growing season, and the mixing ratio of the reused nutrient drainage solution to the fresh one in the closed-loop hydroponic system was maintained within 30% on an EC basis for each irrigation event. The closed hydroponic system had an automated fertigation system and a system for capturing drainage, as well as a system for cleaning up the stored drainage water. The recirculating nutrient solutions were filtered through a sand filter and then sterilized using a UV radiation (ultraviolet, 200–400 nm) disinfection unit manufactured by the NIHHS (Figure 2). The drainage was analyzed once every two weeks during the cultivation period and reused for the rest of the period, except for the three weeks after transplant, when the UV sterilization method could not be applied owing to the high turbidity of the solution. When the Na content in the drainage exceeded 60 ppm, one third of the Na in the drainage tank was discarded. Disposal of sterilized drainage occurred once during the cultivation period in Exp. 1.

### 2.4. Experimental Design

This study consisted of three experiments and three cultivation cycles to develop nutrient composition for paprika cultivation during the circulation of hydroponic cultivation in winter crops. In Exp. 1, the nutrient absorption pattern of paprika according to the growth stage was investigated in the closed coir substrate hydroponic system, with nutrient solution formulations being composed to reflect this characteristic. In Exp. 2, the newly composed nutrient solutions were readjusted, and the hydroponic system's suitability for paprika cultivation was then evaluated by comparing the marketable yield per unit area, photosynthesis, etc., with the existing nutrient solution. During the final cultivation cycle, Exp. 3, the productivity and economic feasibility of the developed nutrient solution (NIHHS-Coir) for closed systems were compared and evaluated with the existing nutrient solution (PBG) for open systems, which is commonly used by farmers.

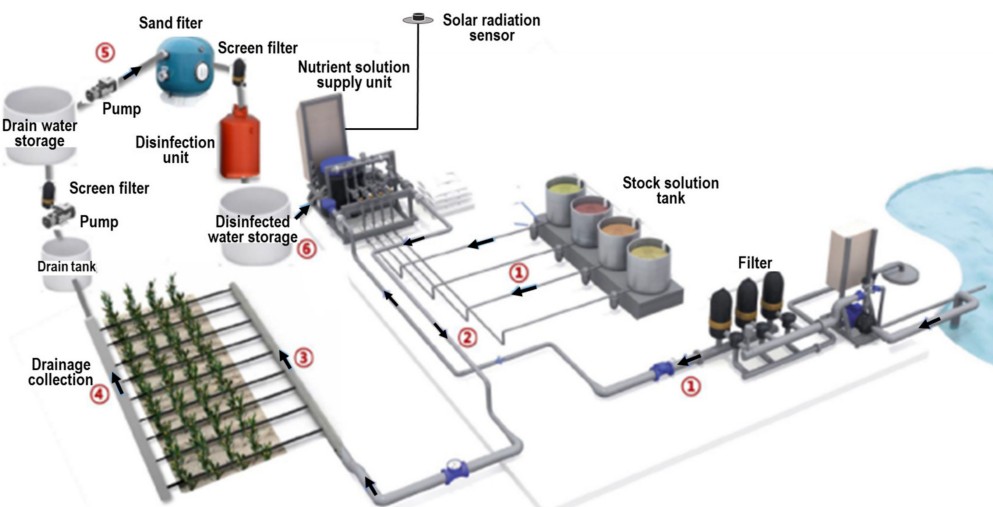

**Figure 2.** A closed hydroponic system consisting of a nutrient solution supply unit to blend fresh water with the recirculated nutrient solution (drainage water), the injection of concentrated stock solutions, and automated irrigation, as well as the filtering and disinfection of stored drainage water. The number indicates the flow of the nutrient solution in the closed hydroponic system.

2.4.1. Nutrient Solution Preparation and Management for Exp. 1

Three kinds of nutrient solutions were used for the treatments (Table 1). The NIHHS nutrient solution was formulated by modifying the PBG nutrient solution [21] considering the nutrient absorption pattern based on the growth stages, reflecting the results of an experiment conducted on the open coir substrate hydroponic system from 2013 to 2014 [22]. The nutrient solution for the closed rockwool cultivation of paprika plants, formulated by the Protected Horticulture Research Station (PHRS), Republic of Korea [23], and the Informatie en Kennis Centrum (IKC) nutrient solution [24] for paprika plants, formulated by the Information and Knowledge Center for Arable Crops and Horticulture, Netherlands, were supplied to each closed hydroponic circuit. Three experimental closed hydroponic circuits for each treatment (three replications) were distributed in a randomized block design in one bay (28 × 16 m) (Figure 1) of the glasshouse; therefore, nine experimental circuits were used for the treatments with one growing gutter (7 m in length, 26 cm in width, and 11 cm in height) placed at a slope of 0.4%. Environmental management in the greenhouse in terms of temperature, humidity, solar radiation, and ventilation, etc., was performed in the same manner as mentioned in Sections 2.1 and 2.3.

**Table 1.** Compositions of the nutrient solutions used for three experimental closed hydroponic circuits for each treatment.

| Nutrient Solution | | $NO_3$-N | $NH_4$-N | P | K | Ca | Mg | $SO_4$-S | Fe |
|---|---|---|---|---|---|---|---|---|---|
| | | (ppm) | | | | | | | |
| NIHHS [z] | Groups [w] 1–2 | 217 | 18 | 37 | 254 | 190 | 36 | 56 | 0.84 |
| | Groups 3–6 | 196 | 16 | 37 | 228 | 152 | 29 | 28 | 0.84 |
| IKC [y] | | 245 | 11 | 42 | 278 | 209 | 46 | 51 | 0.84 |
| PHRS-Rockwool [x] | | 188 | 18 | 40 | 235 | 140 | 28 | 24 | 0.84 |

[z] Formulated by modifying the PBG nutrient solution, considering the nutrient absorption pattern according to growth stages. [y] Formulated by the Informatie en Kennis Centrum (IKC; Information and Knowledge Center for Arable Crops and Horticulture, Netherlands) for paprika cultivation. [x] Formulated by the Protected Horticulture Research Station (PHRS), Republic of Korea, for paprika cultivation in closed rockwool hydroponics. [w] Fruit set period: Groups 1 to 2, October–January; Groups 3 to 6, February–July.

2.4.2. Nutrient Solution Preparation and Management for Exp. 2

To evaluate the suitability of the newly formulated nutrient solution, three types of nutrient solutions were used here: the NIHHS-Coir1 (5–10% readjustment of NIHHS nutrients) and NIHHS-Coir2 (10–15% readjustment of NIHHS nutrients) solutions, readjusted to the concentrations of macronutrients and Fe by considering the nutrient absorption characteristics according to the growth stages, in addition to using the IKC nutrient solution as a control (Table 2). The NIHHS-Coir1 solution was made to lower the concentrations of $NO_3$-N, P, K, Ca, Mg, and $SO_4$-S in the NIHHS solution composition used in Exp. 1 by 5–10% during the growth periods for Groups 1 to 2 and Groups 3 to 6. Meanwhile, the NIHHS-Coir2 solution was prepared by lowering the concentrations of these ions by 10–15% (Table 2).

**Table 2.** NIHHS-Coir1 and NIHHS-Coir2 nutrient solutions used in the experiment to assess the suitability of newly formulated ion ratios for paprika plants in the closed hydroponic system with a coir substrate.

| Nutrient Solution [z] | Groups [y] | $NO_3$-N | $NH_4$-N | P | K | Ca | Mg | $SO_4$-S | Fe |
|---|---|---|---|---|---|---|---|---|---|
| | | (ppm) | | | | | | | |
| NIHHS-Coir1 | 1–2 | 195 | 16 | 37 | 229 | 190 | 33 | 51 | 1.68 |
| | 3–6 | 195 | 16 | 35 | 217 | 171 | 33 | 46 | 1.68 |
| NIHHS-Coir2 | 1–2 | 184 | 15 | 37 | 216 | 171 | 31 | 48 | 1.68 |
| | 3–6 | 186 | 15 | 33 | 205 | 152 | 31 | 41 | 1.68 |

[z] NIHHS-Coir1 (5–10% readjustment of nutrients of NIHHS) and NIHHS-Coir2 (10–15% readjustment of nutrients of NIHHS) nutrient solutions, readjusted to the concentrations of macronutrients and Fe by considering the nutrient absorption characteristics according to growth stages. [y] Fruit set period: Groups 1 to 2, October–January; Groups 3 to 6, February–July.

The nutrient solutions were supplied by proportional control based on the ISR, and the irrigation EC (see Section 2.4.4), frequency, amount of irrigation, and daily irrigation starting and ending points were the same for all treatments. Three experimental closed hydroponic circuits for each treatment (three replications) were distributed in a randomized block design in one bay (28 × 16 m) of the glasshouse, as in Exp. 1. Climate and nutrient solution management in the greenhouse were performed in the same manner as in Exps. 1 and 2.

2.4.3. Nutrient Solution Preparation and Management for Exp. 3

The newly developed nutrient solution for a closed coir substrate hydroponic system, the NIHHS-Coir and existing Netherlands PBG solutions, were tested in closed and open (nonrecirculating) hydroponic systems, respectively. Six experimental closed hydroponic circuits for each treatment (six replications) were distributed in a randomized block design in one bay (28 × 16 m) of the glasshouse, as in Exp. 1. Climate and nutrient solution management in the greenhouse was performed in the same manner as in Exps. 1 and 2.

2.4.4. The pH, EC, and Mixing Ratio of the Reused Nutrient Solution

The nutrient solutions were supplied to all the paprika plants at pH 5.8–6.0 and EC 2.0–3.5 dS·m$^{-1}$ according to the target EC setting for each growth stage: EC 3.0 (dS·m$^{-1}$) (transplant; September); 3.5–3.0 (fruit Groups 1 to 2, October–January); 2.8–2.4 (fruit Groups 3 to 4, February–April); and 2.4–2.0 (fruit Groups 5 to 6, May–July). The mixing ratio of the reused nutrient solution (drainage) to the fresh solution in the closed-loop hydroponic system was maintained within 30% on an EC basis (0.6–1.1 dS·m$^{-1}$) at each irrigation event.

*2.5. Measurements*

2.5.1. Changes in pH, EC, and Mineral Ions in Irrigation, Drainage, and Root Zone

The amounts of irrigation and drainage solution were measured daily, and the pH, EC, and mineral concentrations in the nutrient solutions obtained from the irrigation, drainage, and root zone (RZ) were investigated to analyze the characteristics of the nutrient

absorption patterns of plants based on growth stages in a closed hydroponic system. The nutrient solutions were sampled at one-week intervals and the anions ($NO_3$-N, $PO_4$-P, $SO_4$-S, and $Cl^-$) were analyzed by ion chromatography (DX-500, Dionex, Sunnyvale, CA, USA) and the cations (K, Ca, Mg, and Na) were analyzed by inductively coupled plasma-atomic emission spectrometry (iCAP7400, Thermo Scientific, Waltham, MA, USA).

2.5.2. Yield and Fruit Quality

Paprika plants were pruned to form two main stems that created a vertical 'V' canopy. To investigate the growth characteristics of paprika plants according to the composition of the nutrient solution, plant height, leaf area, and fresh and dry weights of leaves and stems were measured every four weeks during the cultivation period. Commercially ripe (approximately 90%) fruits were harvested twice per week from 10 plants per circuit in 3 replications (30 plants per treatment) to investigate the marketable yield (kg/10a), nonmarketable fruit ratio (%, fruits with cracks, deformations, and physiological disorders such as blossom end rot (EBR)), fruit fresh weight, and total number of fruits per plant. To evaluate fruit quality, the average fruit weight, fruit length and width, number of loculi, flesh thickness, and soluble solids (°Brix) of 15 fruits per treatment were measured. To analyze the soluble solids (°Brix), the fruit was cut in half, the placenta and seeds were removed, approximately 30 g was extracted, the fruit flesh was cut to $0.5 \times 0.5$ cm, and the extract measured using a digital refractometer (PAL-1, Atago Co., Tokyo, Japan).

2.5.3. Photosynthetic Characteristics

Light-response curves for photosynthesis were measured from five plants ($n = 5$) to compare the photosynthetic capacities of paprika plants grown in different nutrient solutions. At the late stage of cultivation (eight months after transplant), $CO_2$ assimilation rate responses to irradiance on the youngest fully expanded leaflets on the fifth leaf of the apical shoot were measured using a photosynthesis measuring apparatus (Li-6400, Li-COR, Lincoln, NE, USA). Photosynthetic light response curves were derived using an internal red/blue LED light source ranging from 0 to 1500 $\mu mol \cdot m^{-2} \cdot s^{-1}$ at a $CO_2$ concentration of 400 $\mu mol \cdot mol^{-1}$, a relative humidity ranging from 50–60%, and a leaf temperature of 25 °C.

*2.6. Statistical Analyses*

Three experimental closed hydroponic circuits with three different nutrient solutions for each treatment in Exps. 1, 2, and 3 were distributed in a randomized block design with three replications. All data were statistically analyzed with the SAS 9.2 (SAS Institute Inc., Cary, NC, USA), and the effects of the different nutrient solution compositions on productivity and fruit quality were evaluated using the analysis of variance (ANOVA). When significant differences were observed, the means were separated using Duncan's multiple range test at a 5% significance level. For significant ($p < 0.05$) effects, the means were separated using the least significant difference (LSD) test at a significance level of 0.05.

**3. Results and Discussion**

*3.1. Changes in pH, EC, and Macronutrients in the Root Zone*

The concentration of $NO_3$-N in the RZ decreased in the fruit set period of Groups 1 and 2, but exceeded 17.0 mM, the recommended concentration of the RZ in closed hydroponic cultivation [25]. During the growth stage, from the fruiting periods of Group 3 (early February) to Group 4 (mid-March), the $NO_3$-N concentration in the RZ did not decrease in all treatments but decreased sharply in the fruit set period of Group 5 (late April to early May) and increased again after the fruit enlargement period (Figure 3a). At 50 days after transplant (DAT), the concentrations of $PO_4$-P in the RZ were below the recommended 1.2 mM [25] in all treatments, but gradually increased from the beginning of the vegetative growth period after the first group's fruit set (Figure 3b). In particular, the $PO_4$-P concentrations in the RZ were higher in the PHRS-Rockwool nutrient solution than in other nutrient solutions, indicating that its use for rockwool cultivation resulted in

an oversupply of PO$_4$-P during closed coir hydroponic cultivation of paprika plants. For K, all the treatment concentrations in the irrigation solution and the RZ were higher than 5.0 mM, the recommended concentration for the RZ [25], accumulating considerably after the fruit set period of Group 3 (Figure 3c).

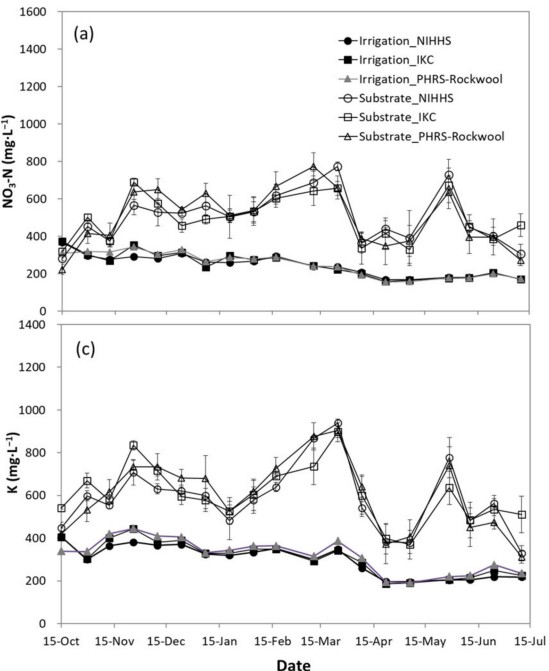
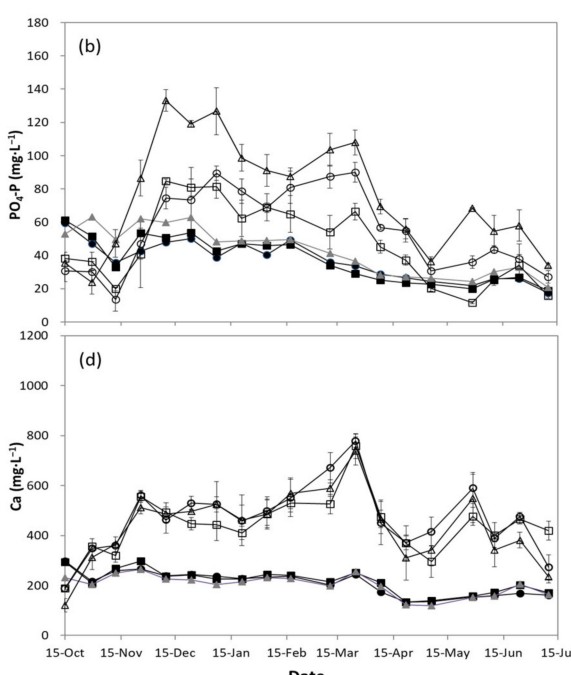

**Figure 3.** Changes in NO$_3$-N (**a**), PO$_4$-P (**b**), K (**c**), and Ca (**d**) in the irrigation and RZ according to the compositions of nutrient solutions NIHHS, IKC, and PHRS-Rockwool during the cultivation period. Vertical bars indicate standard errors of the means (*n* = 3).

Numerous studies on crop nutrient requirements across all growth stages have reported that fruits and vegetables such as paprika, tomato, and melon absorb relatively low nutrients until flowering, but require maximum nutrients during flowering and early fruit development. In fruits and vegetables, nutrient uptake decreases during the growth stages of fruit development and harvest [26,27]. During the fruit set and enlargement period of Group 1, the Ca concentrations in the RZ were below the optimal level (8.5 mM) [25] in all treatments but increased in the subsequent growth stages. All nutrient compositions decreased sharply during the fifth fruit set period; however, Ca concentrations were maintained near the appropriate level during the later growth stages after Group 5 fruiting (Figure 3d).

To prevent problems caused by Ca deficiency, such as blossom end rot (BER), the supply of antagonistic cations such as K, Mg, NH$_4$-N, and Na should be reduced in the early growth stages to facilitate Ca absorption [28,29]. In addition, the RZ EC must be adjusted to an appropriate level by increasing the amount of irrigation supplied during the high-temperature period to ensure that specific cations do not accumulate in the recirculated nutrient solution. The RZ Mg concentrations were higher than the target concentration (4.5 mM, as recommended by de Kreij et al. [25]) during the vegetative growth stage after each group's fruit set was completed in all treatments, particularly before the fruit set of Group 5 (Figure 4a). In the late growth stage, the cation concentrations of K, Ca, and Mg in the RZ were higher in the treatment with the IKC nutrient solution than in other compositions. The concentration of SO$_4$-S in the RZ exceeded the recommended concentration of 3 mM [25] in all treatments, particularly during the initial fruit growth stage in Groups 1 and 2 in the NIHHS and IKC (Figure 4b).

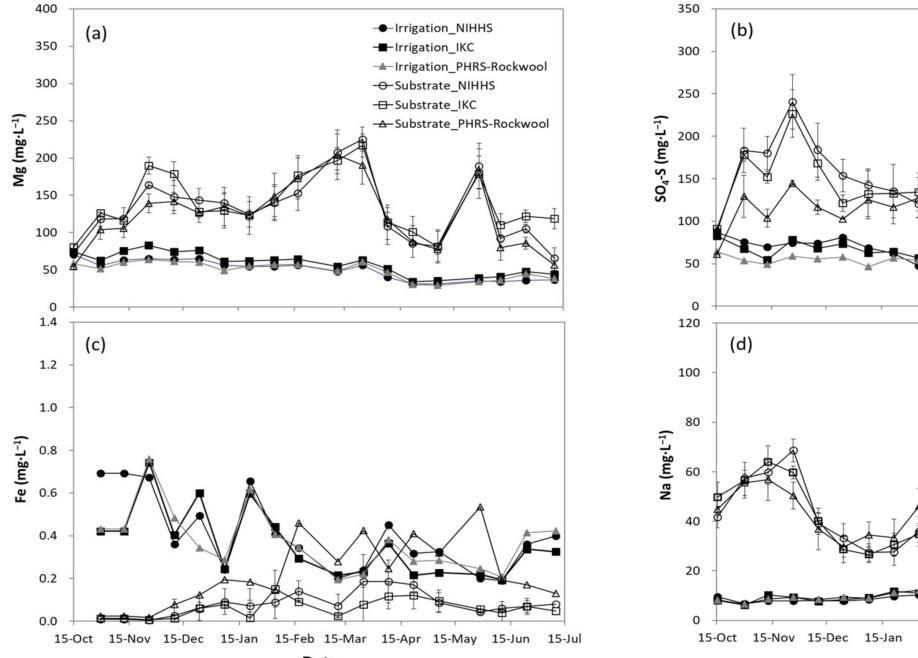
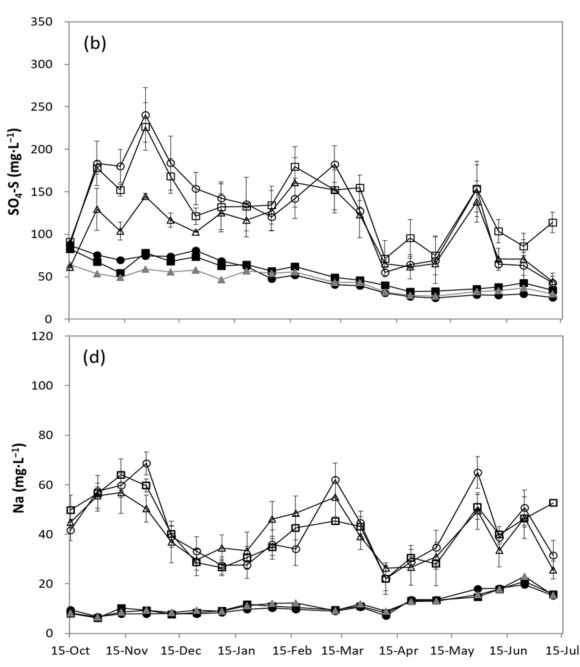

**Figure 4.** Changes in Mg (**a**), SO$_4$-S (**b**), Fe (**c**), and Na (**d**) in the irrigation and RZ according to the compositions of the nutrient solutions: NIHHS, IKC, and PHRS-Rockwool, during the cultivation period. Vertical bars indicate standard errors of the means (*n* = 3).

The concentrations of Fe in the RZ were considerably lower than the appropriate concentration level (0.84 ppm) for all nutrient solution compositions throughout the cultivation period (Figure 4c), thus necessitating the increased concentration of Fe chelates in the irrigation nutrient solution or providing additional supplies during the cultivation period. The Cl concentration in the RZ was high from the transplant to the fruit-enlargement period of Group 1—reflecting the Cl ions released from the substrate—and stabilized below 50 ppm after the reproductive stage of Group 2. In the presence of salts such as Na and Cl in irrigation water, salts leach from the substrates using excess water [30], thus wasting both water and fertilizer [31]. Therefore, the nutrient solution must be recirculated for long periods and periodically discharged whenever the salt concentration exceeds the acceptable threshold level [5]. Similar to Cl, Na in the RZ was continuously discharged from the substrate, but the Na and Cl concentrations in the drainage solution were monitored and managed; they could therefore be maintained below the allowable value of 60 ppm for most of the cultivation period in all treatments (Figure 4d).

The pH in the RZ gradually increased over time within the range of pH 6.5 to 7.3 in all treatments from 15 DAT and decreased again 50 DAT, maintaining a pH range of 6.0 to 6.2. The increase in pH in the RZ and drainage solution in the early stages was similar to that observed in previous experiments, possibly owing to the increased absorption of anions during the early growth stages of paprika in closed hydroponic systems [32,33]. During the fruiting period in Group 1, the EC in the RZ in the NIHHS and IKC nutrient solution treatments increased to the target EC of 5–6 dS·m$^{-1}$, whereas the EC in the treatment of the PHRS-Rockwool nutrient solution was 1 dS·m$^{-1}$ lower. In the growth stage after the fruiting period of Group 3, the pH in the RZ was generally highest in the IKC solution.

For Fe, there was a large difference between the ion concentrations in the irrigation nutrient solution and in the substrate throughout the growth period. The differences between the ion concentrations in the RZ and with irrigation showed a negative value; therefore, the Fe concentration in the nutrient solution must be increased to maintain the appropriate Fe concentration in the RZ. In sterilization by UV radiation (200–400 nm) in a closed hydroponic system, the iron chelate in the nutrient solution is photooxidized and

precipitated, thus reducing the content of iron available for absorption by plants. Therefore, the Fe content should be corrected through additional supply [34].

The differences between the $NO_3$-N, K, Ca, and Mg concentrations in RZ and with irrigation solution increased considerably in Groups 3 and 4 compared to Groups 1 and 2. By lowering the concentration during irrigation after the fruit set period for Group 3, ion adjustment in the nutrient solution, reflecting the nutrient absorption characteristics of paprika, was found to be necessary. Throughout the cultivation period, the differences between $PO_4$-P concentrations in the irrigation and RZ for the NIHHS and IKC nutrient solutions were lower than those of the PHRS-Rockwool nutrient solution, increasing after Group 3 and decreasing again after Group 5. To create a nutrient solution for each growth stage, it was necessary to maintain the proper concentration in the RZ by making the concentration of $PO_4$-P the same as the original composition in Groups 1 and 2 and lowering the irrigation concentration after Group 3. Throughout the growth period, the differences between $SO_4$-S concentrations in irrigation solution and RZ were higher than the irrigation concentration owing to the magnesium sulfate or potassium sulfate fertilizers used to match Mg and K supplies in the nutrient solution composition. The Bar-Yosef study found that increasing the EC of irrigation water from $2.3 \pm 0.2$ to $4.0 \pm 0.6$ $dSm^{-1}$ significantly reduced the N concentration of diagnostic leaves and increased Ca and Mg concentrations, resulting from their accumulation in recycled solutions [35]. In this study, accumulation of K, Ca, and Mg was observed during the fruiting period of Group 4, with high EC in the RZ.

### 3.2. Paprika Productivity and Quality Based on the Nutrient Solution Compositions

In the closed hydroponic system with a coir substrate, the fruit yield and quality characteristics of paprika according to the growth stage were analyzed for each nutrient solution composition (Table 3). In Groups 1 and 2, the marketable yield (kg/10a) and fruit weight per plant were high in the NIHHS and NIHHS-Rockwool nutrient solutions. In Groups 5 and beyond, the marketable yield, marketable fruit ratio (%), and fruit weight per plant were high in the NIHHS and IKC nutrient solutions. As a result, the treatment with NIHHS nutrient solution throughout the growth period showed stable productivity. The properties of paprika fruit quality showed statistically significant differences in average fruit weight and soluble solid content according to the nutrient solution composition and high significance in average fruit weight, flesh thickness, and soluble solids according to the growth stages. The soluble solids (°Brix) in the nutrient solution compositions were lowest in the IKC cultures, with the exception of Group 6, and showed higher values in NIHHS and NIHHS-Rockwool nutrient solutions. Several studies have shown that horticultural crops grown in closed hydroponic systems exhibit reduced vitality and growth over time compared to those grown in open systems [35,36]. The decrease in marketable yield and quality may be due to mineral imbalances or changes in nutrient solution concentrations. Furthermore, pH may change the availabilities of certain essential nutrients [37].

### 3.3. The pH and EC of the Newly Formulated Nutrient Solution

The composition of mineral ions in the NIHHS nutrient solution was readjusted to the NIHHS-Coir1 (5–10% readjustment of nutrients in NIHHS) and NIHHS-Coir2 (10–15% readjustment of nutrients in NIHHS) through analysis of the Δ ion values and the nutrient absorption characteristics of paprika plants according to the growth stage (Table 2). There were no significant differences between treatments in the daily drainage ratio (%) of each nutrient solution within the range of 10–40%.

The RZ pH was slightly higher than that of the irrigation in all nutrient solution compositions up to 70 DAT, and the IKC nutrient solution in the growth stage of Groups 1 and 2 showed a higher RZ pH than the other two nutrient solution compositions (Figure 5). The EC of the irrigation was supplied from 3.0 to 3.3–3.5 $dS·m^{-1}$ to render the target concentration of RZ 5.0 $dS·m^{-1}$ in the fruit set period of Groups 1 and 2 with high nutrient requirements. During the fruit set period of Groups 1 and 2, the RZ EC of the NIHHS-Coir1 and IKC nutrient solutions increased from 5 to 6 $dS·m^{-1}$, but that in the

NIHHS-Coir2 nutrient solution was lower. The RZ EC was highest in the IKC nutrient solution and lowest in the NIHHS-Coir2 nutrient solution (Figure 5).

**Table 3.** Marketable yield characteristics of paprika 'Cupra' affected by the compositions of the nutrient solutions, NIHHS, IKC, and PHRS-Rockwool in Experiment 1, according to growth stages during the winter cropping season in a closed hydroponic system with a coir substrate.

| | | Marketable Yield (kg/m$^2$) | Marketable Fruit Ratio (%) | Fruit Weight (kg)/Plant ($n$ = 30) | No. of Fruits/Plant ($n$ = 30) |
|---|---|---|---|---|---|
| Groups (G) [x] | | | | | |
| 1–2 | | 4.95 c [z] | 98.1 a | 1.45 c | 7.9 c |
| 3–4 | | 7.10 a | 97.1 a | 2.09 a | 13.2 b |
| 5–6 | | 5.60 b | 90.6 b | 1.76 b | 17.8 a |
| Nutrient solutions (NS) | | | | | |
| NIHHS | | 6.03 a | 96.2 a | 1.79 | 13.2 |
| IKC | | 5.97 a | 96.4 a | 1.77 | 13.2 |
| NIHHS-Rockwool | | 5.65 b | 93.2 b | 1.74 | 12.4 |
| G × NS | | | | | |
| 1–2 | NIHHS | 5.06 d | 98.3 a | 1.47 | 7.8 |
| | IKC | 4.78 e | 98.7 a | 1.38 | 7.3 |
| | NIHHS-Rockwool | 5.02 d | 97.2 ab | 1.50 | 8.5 |
| 3–4 | NIHHS | 7.14 a | 97.1 ab | 2.10 | 13.6 |
| | IKC | 7.21 a | 98.3 a | 2.09 | 12.2 |
| | NIHHS-Rockwool | 6.96 b | 95.9 b | 2.09 | 13.9 |
| 5–6 | NIHHS | 5.88 c | 93.0 c | 1.81 | 18.0 |
| | IKC | 5.93 c | 92.2 c | 1.84 | 17.8 |
| | NIHHS-Rockwool | 4.97 d | 86.5 d | 1.64 | 17.6 |
| Significance [y] | | | | | |
| G | | *** | *** | *** | *** |
| NS | | *** | *** | NS | NS |
| G × NS | | *** | ** | NS | NS |

[z] The mean values followed by the different lowercase letters in each column indicate significant differences at $p \leq 0.05$ using the LSD test. [y] NS indicates not significant and **, *** represent significant at $p \leq 0.01$, or 0.001, respectively, by two-way ANOVA. [x] Fruit set period: Groups 1 to 2, October–January; Groups 3 to 4, February–April; Groups 5 to 6, May–July.

In the growth stages after Group 3, the RZ pH was highest under IKC nutrient solution treatment. The inequivalent uptake of cations by plants, as well as the ratio of nitrate to ammonium in the irrigation nutrient solution, affect pH changes in the root environment [38]. In addition, Reshef et al., who investigated the effect of threshold EC on paprika yields in a closed-irrigation loop system, reported that an increase in threshold EC resulted in a reduction in marketable fruit weight, yield decline, and a decrease in relative fruit weight of 5.3% per dS·m$^{-1}$ [35]. As the temperature inside the greenhouse gradually increased after Group 3, the RZ EC increased for all nutrient solution compositions; particularly, the IKC nutrient solution EC was higher than those of the NIHHS-Coir1 and NIHHS-Coir2 nutrient solutions. After the fruit set period for Group 3, the irrigation was supplied at a lower EC concentration of 2.8 dS·m$^{-1}$, and to increase the moisture content in the substrate, the total irrigation frequency was increased and the irrigation amount per serving decreased by lowering the set value of the integrated solar radiation. During the high-temperature period after May, the RZ EC was highest in the IKC nutrient solution and lowest in the NIHHS-Coir2 nutrient solution. Throughout the growth period, the NIHHS-Coir1 nutrient solution showed more stable RZ EC changes compared with the other two.

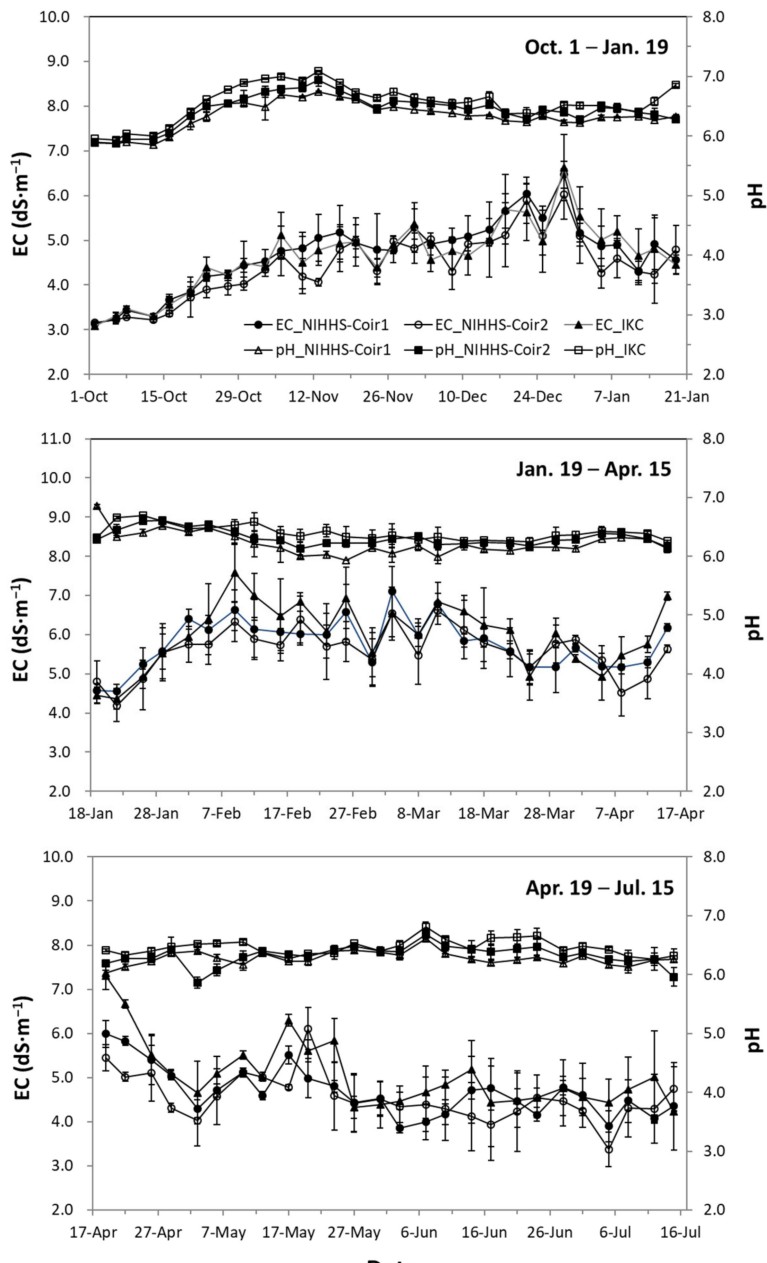

**Figure 5.** Changes in EC and pH in the RZ of each nutrient solution composition (NIHHS-Coir1, NIHHS-Coir2, and IKC) in the closed hydroponic system with a coir substrate. Vertical bars indicate standard errors of the means (*n* = 3).

### 3.4. Suitability Evaluation of the Newly Formulated Nutrient Solution

Analysis of the concentration changes of mineral ions in the irrigation and RZ according to the growth stage showed that the $NO_3$-N concentration in the RZ exceeded the optimal concentration set for rockwool hydroponics in all nutrient solution compositions (Figure 6). Coconut coir is an organic medium with a high cation exchange capacity (CEC) ranging from 320 to 950 $mmol_c$ $kg^{-1}$, and it has a strong ability to adsorb nutrients due to its buffering capacity [19]. Therefore, application of the optimal ion concentration level in the RZ based on rockwool, an inorganic medium, to a hydroponic system using a coir substrate is not suitable. The RZ concentrations of $NO_3$-N decreased in all nutrient solution compositions from mid-March, the fruit set period for Group 4, owing to the lower EC irrigation level. The RZ concentration of $PO_4$-P in the IKC nutrient solution was considerably lower than 1.2 mM, an appropriate level as stated by de Kreij et al. [25] until the fruit

set period for Group 3. However, the NIHHS-Coir1 and NIHHS-Coir2 nutrient solutions formulated by reflecting the $PO_4$-P absorption characteristics of paprika plants exceeded the appropriate RZ concentration level (Figure 6).

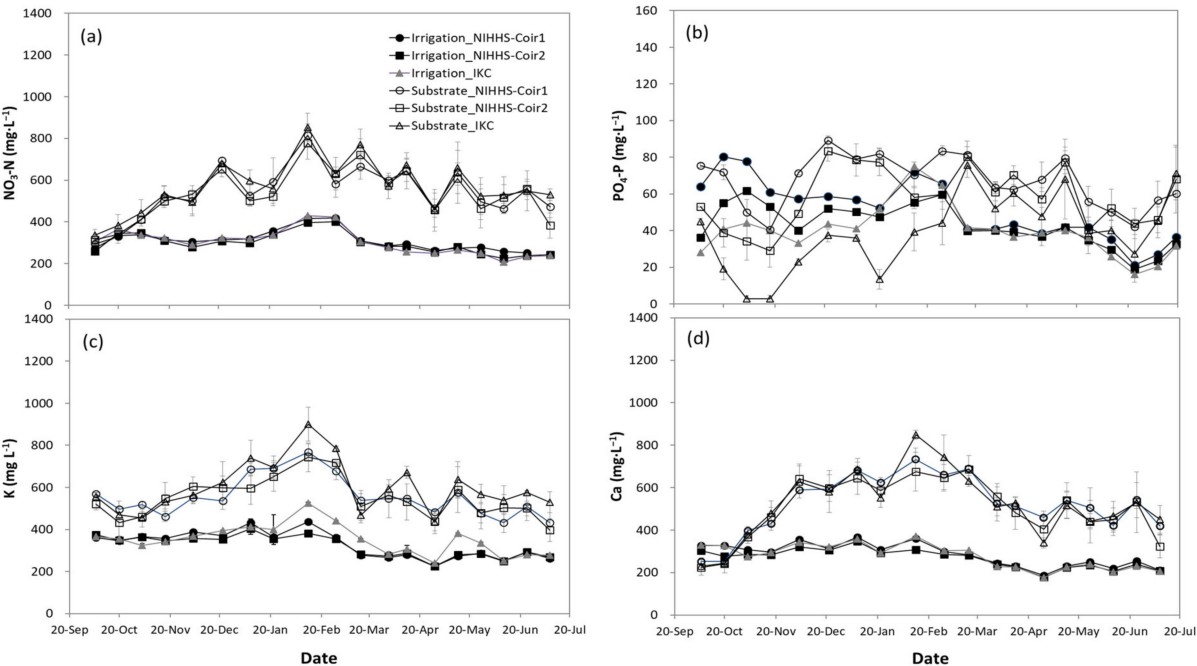

**Figure 6.** Changes in $NO_3$-N (**a**), $PO_4$-P (**b**), K (**c**), and Ca (**d**) in the irrigation and RZ of each nutrient solution (NIHHS-Coir1, NIHHS-Coir2, and IKC) in the closed hydroponic system with a coir substrate. Vertical bars indicate standard errors of the means ($n = 3$).

Although the NIHHS-Coir1 and NIHHS-Coir2 nutrient solutions did not show a decrease in RZ $PO_4$-P in the early growth stages, the ratio of $PO_4$-P in the nutrient solution was finally reduced by 5% because the RZ accumulation appeared after the fruit set period of Group 2. In Experiment 1, the K concentration exceeded the appropriate RZ concentration level [25] for all compositions. The concentration tended to increase significantly from the fruit set period of Group 4, but the RZ K concentrations of the NIHHS-Coir1 and NIHHS-Coir2 nutrient solutions were significantly reduced compared to those of the IKC solution (Figure 6). In the early stages of growth, the NIHHS-Coir1 and NIHHS-Coir2 solutions, which reduced the supply of K and increased the ratio of Ca, did not show a rapid increase in RZ Ca, as in Experiment 1 after Group 3. In addition, the NIHHS-Coir1 solution did not show a rapid decrease in Ca compared to other compositions during the fruit set period of Group 5. Ropokis et al. [39] reported that for paprika cultivation in a closed hydroponic system, when the Ca concentration in the irrigation solution was supplied at 3.0 me·$L^{-1}$, Ca accumulation in the recirculating nutrient solution did not occur; however, at 6.0, 9.0, and 12.0 me·$L^{-1}$, it increased to 34, 56, and 74 me $L^{-1}$ by the later cropping stages, respectively, whereas EC increased to 6.4, 9.0, and 10.8 dS·$m^{-1}$, respectively. However, in this study, RZ Ca accumulation was not observed in the newly formulated NIHHS-Coir1 and NIHHS-Coir2 solutions, with the exception of the IKC solution at a high concentration, as seen in the results obtained by Ropokis et al. [39]. This could be attributed to the balanced composition ratio with other major nutrients, maintained at an appropriate EC according to the growth stage, even though the Ca concentrations in the NIHHS-Coir1 and NIHHS-Coir2 solutions in the later cropping stages were 8.6 and 7.6 me·$L^{-1}$, respectively.

The RZ Mg concentration exceeded the appropriate level throughout the cultivation period for all treatments (Figure 7). In Experiment 2, the accumulation of salt ions such as K, Ca, and Mg in the RZ was larger in the IKC solution, which increased significantly after the fruit set period in Group 3. This necessitates increased irrigation during high

temperatures ($\geq$30 °C) to adjust the concentration within the RZ. The RZ concentration of $SO_4$-S also exceeded the appropriate level for cultivation periods, particularly in the IKC solution, which showed a higher concentration than those in other solutions after the fruit set period for Group 3. During nutrient solution recycling, the Cl and Na ions were continuously leached from the substrate but remained within the permissible level of 60 ppm or less within the RZ, and there was no disposal of the recirculated drainage solution. The RZ concentration of Fe was below the appropriate concentration level set by de Kreij et al. [25] for all nutrient solution compositions throughout the cultivation period. The NIHHS-Coir1 and NIHHS-Coir2 solutions, which increased the concentration of Fe in the irrigation, maintained a higher RZ concentration of Fe than in the IKC solution.

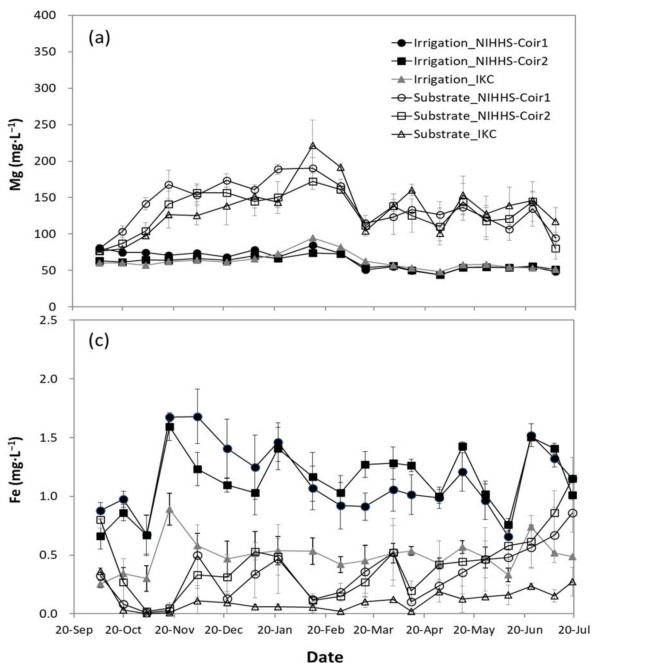
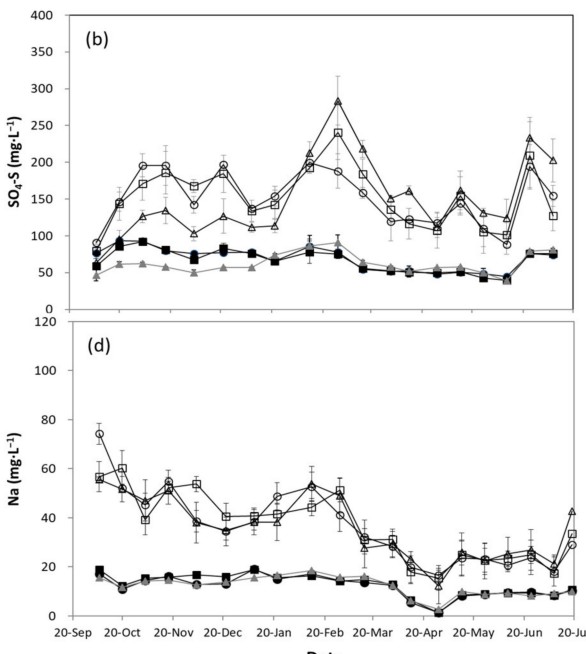

**Figure 7.** Changes in Mg (**a**), $SO_4$-S (**b**), Fe (**c**), and Na (**d**) in the irrigation and RZ of each nutrient solution composition (NIHHS-Coir1, NIHHS-Coir2, and IKC) in the closed hydroponic system with a coir substrate. Vertical bars indicate standard errors of the means (*n* = 3).

Changes in the concentrations of cations such as K, Ca, and Mg in the nutrient solutions obtained from the irrigation and RZ in the three experimental closed hydroponic circuits were analyzed during the cultivation period. The K/Ca ratios in the irrigation and RZ of the IKC solution were low in the early stages of growth (the fruiting period of Group 1), but were later constantly higher than those of the other solutions. The NIHHS-Coir2 nutrient solution had a lower RZ K/Ca ratio than the other two compositions. During the cultivation period, the irrigation K/Mg ratio was highest in the IKC solution, and the RZ K/Mg ratio was higher until the fruit set period for Group 4 compared to the other two compositions. This pattern was also shown in K/(Ca + Mg), and the ratios of these cations such as K/Ca and K/Mg in the irrigation and RZ were lower in the nutrient solution of NIHHS-Coir1, particularly after the middle of cultivation, compared with the other two compositions (Figure 8).

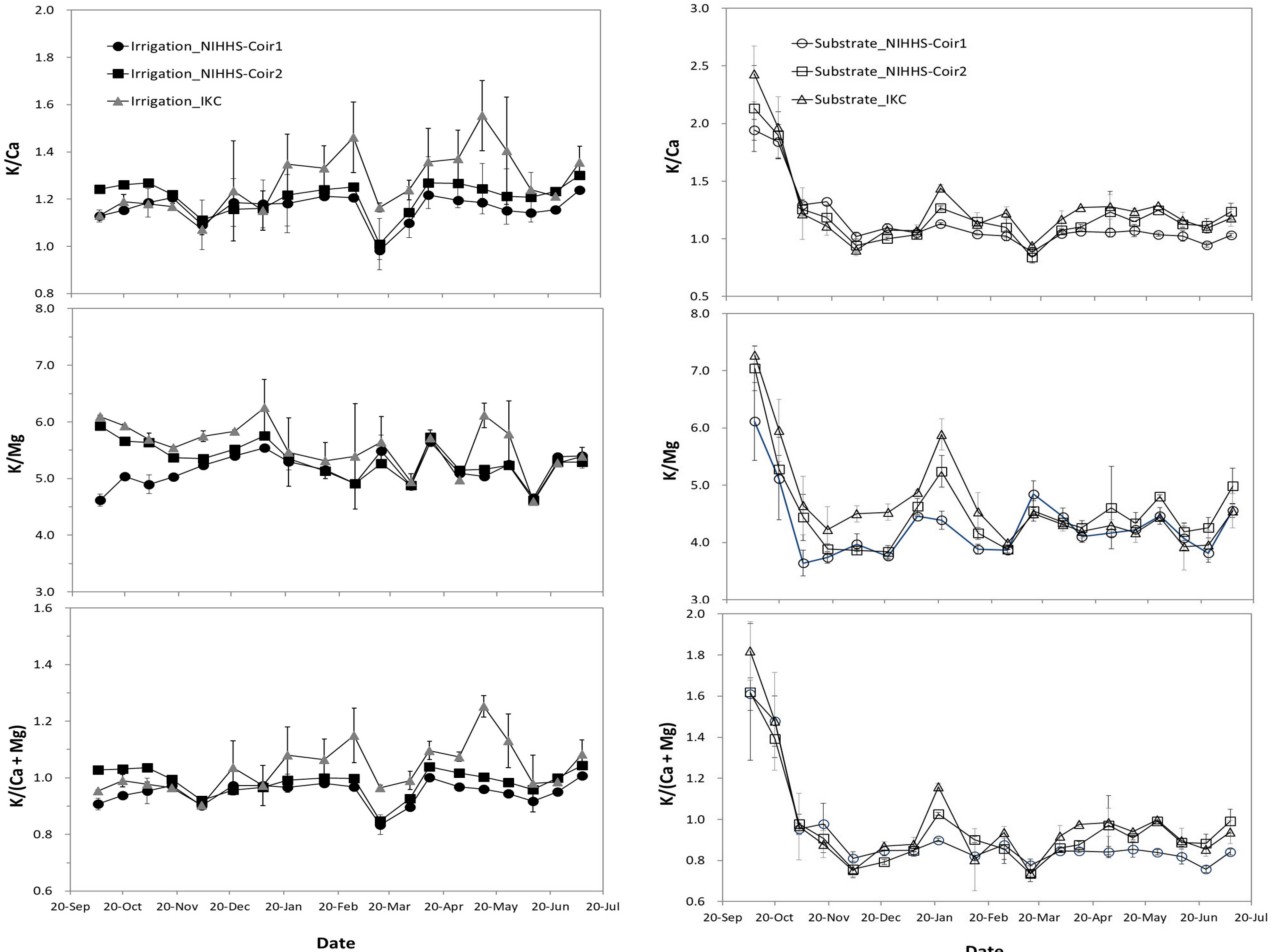

**Figure 8.** Changes in K/Ca, K/Mg, and K/(Ca + Mg) in the irrigation and RZ of each nutrient solution composition (NIHHS-Coir1, NIHHS-Coir2, and IKC) in the closed hydroponic system with a coir substrate. Vertical bars indicate standard errors of the means (*n* = 3).

*3.5. Photosynthetic Responses, Yield, and Fruit Characteristics of Paprika*

The photosynthetic rates in the fully expanded paprika leaves by irradiance at the late stage of cultivation (eight months after transplant) as affected by different nutrient solution compositions were found to be slightly higher in the NIHHS-Coir1 nutrient solution than in the NIHHS-Coir2 and IKC nutrient solutions (Figure 9). Several studies have shown that photosynthesis depends on mineral nutrition because numerous nutrients are involved in plant metabolism and the electron transport chain [40]. P, Mg, K, and Ca are involved in the synthesis of phosphorylated metabolites and chlorophyll, the photoreaction of thylakoid membranes, and electron transport to improve photosynthesis efficiency and pore conductivity [41]. Analysis of the $CO_2$ fixation rate of paprika leaves according to the nutrient solution composition during the late growth stage showed no significant difference between the compositions at photosynthetically active radiations (PARs) below 1000 $\mu mol \cdot m^{-2} s^{-1}$, while PARs above 1200 $\mu mol \cdot m^{-2} s^{-1}$ showed increased photosynthesis in the NIHHS-Coir1 solution. In addition, the photosynthetic rates at high PARs of 1200 $\mu mol \cdot m^{-2} s^{-1}$ or more showed no significant difference between NIHHS-Coir2 and IKC nutrient solutions (Figure 9).

Throughout the cultivation period, the NIHHS-Coir1 solution had a higher marketable fruit ratio of 2.4 to 3.5% and a higher marketable yield (kg/10a) of 5.3 to 8.7% than those of the IKC solution. The NIHHS-Coir1 solution showed a stable yield and fruit quality throughout the cultivation period, with a marketable fruit ratio ranging from 2.4 to 4.9%



and a marketable yield ranging from 10.8 to 14.4% higher than the NIHHS-Coir2 solution after fruiting in Group 3 (Table 4).

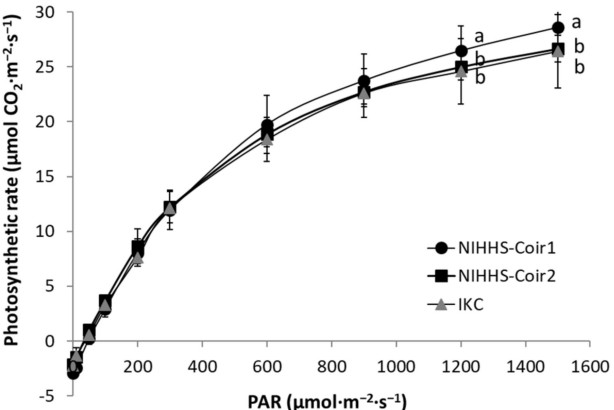

**Figure 9.** Light-response curves of net photosynthesis in the fully expanded leaves of paprika 'Cupra' at the late stage of cultivation (eight months after transplant) as affected by different nutrient solution compositions in the closed hydroponic system with a coir substrate. DMRT at $p \leq 0.05$ and different lowercase letters indicate significant differences between treatments. Vertical bars indicate standard errors of the means ($n = 5$).

**Table 4.** Marketable yield characteristics of paprika 'Cupra' affected by the compositions of the nutrient solutions in Experiment 2, NIHHS-Coir1, NIHHS-Coir2, and IKC, according to growth stages during the winter cropping season in a closed hydroponic system with a coir substrate.

| | | Marketable Yield (kg/m²) | Marketable Fruit Ratio (%) | Fruit Weight (kg)/Plant ($n = 30$) | No. of Fruits/Plant ($n = 30$) |
|---|---|---|---|---|---|
| Groups (G) [x] | | | | | |
| 1–2 | | 4.26 a [z] | 99.4 a | 1.22 b | 6.7 c |
| 3–4 | | 4.03 b | 89.8 b | 1.30 ab | 9.4 b |
| 5–6 | | 4.07 b | 80.6 c | 1.45 a | 14.3 a |
| Nutrient solutions (NS) | | | | | |
| NIHHS-Coir1 | | 4.30 a | 92.4 a | 1.36 | 10.3 |
| NIHHS-Coir2 | | 3.96 b | 88.3 b | 1.29 | 9.9 |
| IKC | | 4.10 b | 89.0 b | 1.33 | 10.2 |
| G × NS | | | | | |
| | NIHHS-Coir1 | 4.39 a | 99.8 | 1.26 | 6.8 |
| 1–2 | NIHHS-Coir2 | 4.35 ab | 99.1 | 1.25 | 6.9 |
| | IKC | 4.04 c | 99.4 | 1.16 | 6.3 |
| | NIHHS-Coir1 | 4.17 abc | 93.8 | 1.32 | 9.4 |
| 3–4 | NIHHS-Coir2 | 3.76 d | 88.1 | 1.22 | 8.9 |
| | IKC | 4.16 abc | 87.6 | 1.36 | 9.7 |
| | NIHHS-Coir1 | 4.33 ab | 83.6 | 1.50 | 14.6 |
| 5–6 | NIHHS-Coir2 | 3.78 d | 77.7 | 1.39 | 13.9 |
| | IKC | 4.11 bc | 80.2 | 1.46 | 14.4 |
| Significance [y] | | | | | |
| G | | ** | *** | * | *** |
| NS | | *** | ** | NS | NS |
| G × NS | | ** | NS | NS | NS |

[z] The mean values followed by the different lowercase letters in each column indicate significant differences at $p \leq 0.05$ using the LSD test. [y] NS indicates not significant and *, **, *** represent significant at p≤0.05, 0.01 or 0.001, respectively, by two-way ANOVA. [x] Fruit set period: Groups 1 to 2, October–January; Groups 3 to 4, February–April; Groups 5 to 6, May–July.

The ion concentrations of $NO_3$-N, P, Ca, and $SO_4$-S in the NIHHS-Coir1 solution were slightly readjusted by analyzing the concentration of each RZ ion according to growth stages.

Finally, the nutrient solution formulation suitable for cultivating paprika in closed hydroponic systems with a coir substrate in the winter cropping season (August of the previous year to July) in Republic of Korea was formulated by dividing the growth stage into 'Groups 1 to 2' and 'Group 3 and beyond,' and by readjusting the concentrations of $NO_3$-N, $NH_4$-N, P, K, Ca, Mg, $SO_4$-S, and Fe as shown in Table 5. To evaluate the productivity of paprika grown in the newly formulated nutrient solution in a closed hydroponic system, the NIHHS-Coir 1 and existing Netherlands PBG solutions were tested in closed and open (nonrecirculating) hydroponic systems, respectively. A comparison of productivity according to growth stage in each system revealed no significant differences in the marketable yield and fruit ratio. In addition, paprika plants grown in the NIHHS-Coir1 solution showed a lower incidence of BER than those grown in open systems using the PBG solution. Lee et al. [29] reported a strong correlation (r = 0.82) between the proportion of K in the PBG solution and the occurrence of BER during paprika hydroponics. The ratio of $K^+$ to the total cations of the NIHHS-Coir1 solution was lower than that of the PBG solution, resulting in a decrease in the incidence of BER. After analysis of the water and fertilizer savings from closed coir substrate hydroponics in comparison with the conventional cultivation method (open-system) in Republic of Korea, it was estimated that when paprika was grown at the recommended planting density (6.6–7.4 stems/m$^2$) for 300 d, groundwater and fertilizer usage were reduced by 3609 tons/ha and 7218 kg/ha, respectively.

**Table 5.** Newly developed nutrient solution compositions for paprika cultivation in the closed hydroponic system with a coir substrate in the winter cropping season.

| Nutrient Solution | Groups | NO$_3$-N | NH$_4$-N | P | K | Ca | Mg | SO$_4$-S | Fe | K/Ca Ratio |
|---|---|---|---|---|---|---|---|---|---|---|
| | | (ppm) | | | | | | | | |
| NIHHS-Coir (closed) | 1–2 | 200 | 13 | 35 | 227 | 190 | 33 | 51 | 1.68 | 1.19 |
| | 3–6 | 190 | 15 | 33 | 217 | 162 | 33 | 43 | 1.68 | 1.33 |

## 4. Conclusions

Closed hydroponics are effective in environmental conservation and operation cost reduction by collecting discarded drainage, disinfecting and adjusting it, and supplying it back to crops. In Korea, 90% of hydroponic farms have problems with environmental pollution and unnecessary use of fertilizer because open hydroponics residual fertilizer components are discharged from the greenhouse. Korean hydroponic farms have recently switched to ecofriendly organic substrates, such as coconut coir, an easily recycled substrate. Therefore, the closed hydroponic system is expected to be rapidly distributed from large-scale export horticultural complexes to general farms considering environmental sustainability. Paprika growers in Korea are currently following European nutrient solution management technologies for different climates and cropping seasons, and a nutritional solution composition for paprika cultivation has not been developed to suit the cultivation characteristics of Northeast Asia. This study investigated the nutrient absorption pattern according to growth stages and determined the optimal nutrient solution composition suitable for cultivating paprika (*Capsicum annum* L.) in closed hydroponic systems with a coir substrate during the winter cropping season in Republic of Korea. To reduce the nutritional imbalances in the concentration ratio of major nutrients in the RZ, the nutrient solution in the closed coir substrate hydroponic system was formulated with new nutrient compositions according to growth stages, considering nutrition absorption patterns, ion balances between anions and cations, and Δ ion concentration. Three nutrient solutions were used to evaluate its suitability. Throughout the entire cultivation period, the newly formulated NIHHS-Coir1 solution had a marketable fruit ratio (%) of 2.4–3.5% higher and a marketable yield (kg/m$^2$) of 5.3–8.7% higher than those of the conventional IKC solution.

In the closed hydroponic cultivation system with a coir substrate for winter cropping cultivation, the nutrient solution composition suitable for growing paprika was formulated

by dividing the growth stages into Groups 1 to 2 and Groups 3 to 6. The economic benefit through the increased marketable yield as a result of the application of the newly developed nutrient solution was estimated to be up to approximately $15,350/ha, compared with the existing nutrient solution.

In addition, the efficacy of this system was evaluated by comparing the paprika productivity and economic feasibility it yielded with those of paprika grown in open hydroponic systems through a field application test; there was no significant difference between hydroponic methods in terms of marketable yield, and the occurrence of BER was reduced. After analyzing the water and fertilizer savings as a result of closed coir substrate hydroponics, it was estimated that the uses of groundwater and fertilizer per cultivation cycle decreased by 3609 t/ha and 7218 kg/ha, respectively, compared with the open system. These results are expected to contribute to the distribution of environmentally friendly closed hydroponic systems suitable for paprika cultivation in Korea.

**Author Contributions:** Conceptualization, methodology, investigation, writing—original draft, editing, K.-H.Y.; conceptualization, supervision, validation, review and editing, K.-Y.C.; conceptualization, review and editing, G.-L.C., J.-H.L. and K.-S.P. All authors have read and agreed to the published version of the manuscript.

**Funding:** This research was financially supported by RDA, Republic of Korea, grant numbers PJ01047901 and PJ01608501. The funders had no role in the design of: the study; the collection, analysis, or interpretation of data; the writing of the manuscript; or the decision to publish the results.

**Data Availability Statement:** Not applicable.

**Conflicts of Interest:** The authors declare no conflict of interest.

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
