# Peer review of "Development of Nutrient Solution Compositions for Paprika Cultivation in a Closed Coir Substrate Hydroponic System in Republic of Korea’s Winter Cropping Season"

_horticulturae, doi:10.3390/horticulturae9040412_

Round 1

Reviewer 1 Report

In this article, the nutrient solution is formulated suitable for cultivating paprika in closed hydroponic systems with a coir substrate during winter crop cultivation. The problem is well explained in the introduction and the objectives are clear. However, there were some suggestions to revise before the article is published.

-What is RZ in the abstract? Root zone? When the term is used first time in the text should be explained in brackets.

-It is suggested to add figures or schemes with explanations in the text of how the Closed Coir-Substrate Hydroponic System works. (2.1. Plant materials and experimental conditions)

-What is UV? Line 119 It is suggested to give an explanation of when the term is first time met in the text.

-"Nutrient absorption characteristics of paprika ‘Cupra’ with growth stages in the hydroponic system with a coconut coir substrate". If this article is written in Korean language, then please add at the end of the reference the language in which the article was written.

Author Response

Response to Reviewer 1 Comments

Thank you for reviewing the manuscript (Development of Nutrient Solution Compositions for Paprika Cultivation in a Closed Coir-Substrate Hydroponic System in South Korea’s Winter Cropping Season) and for helpful suggestions. We revised the manuscript, and have submitted the revised one. Note that references to lines are with regard to the manuscript as it was received, and these have shifted as changes were made.

Point 1: (Line 24) What is RZ in the abstract? Root zone? When the term is used first time in the text should be explained in brackets.

Response 1: The RZ in the abstract has been replaced with ‘root zone (RZ)’ (line 28).

Point 2: It is suggested to add figures or schemes with explanations in the text of how the Closed Coir-Substrate Hydroponic System works. (2.1. Plant materials and experimental conditions)

Response 2: Explanations have been added to the text with an illustration of how the closed coir-substrate hydroponic system works in the section ‘2. Closed Hydroponics System Setup’ (Fig. 1, Fig. 2, line 163-188).

Point 3: (Line 119) What is UV? It is suggested to give an explanation of when the term is first time met in the text.

Response 3: A description of UV and its explanation has been added to the text (line 174-178, line 393-395).

Point 4: (Line 616) "Nutrient absorption characteristics of paprika ‘Cupra’ with growth stages in the hydroponic system with a coconut coir substrate". If this article is written in Korean language, then please add at the end of the reference the language in which the article was written.

Response 4: The reference "Nutrient absorption characteristics of paprika ‘Cupra’ with growth stages in the hydroponic system with a coconut coir substrate" has been corrected (line 738).

Reviewer 2 Report

ABSTRACT:

The abstract must have a rationale, an objective, materials and methods, results, and conclusions.

Lines 16-18. Please Re-write the first sentence must be a rationale and please write a problem statement for this study.

Lines 18-21. The objective of the study not clear. Please write in one paragraph the main objectives.

Lines 21-23. Please do not use abbreviation in the abstract such as KC, and PHRS.

The authors should mention the treatments and experimental design to explain the main findings.

Please insert the main results for the whole study.

INTRODUCTION:
The introduction section is relatively short and very good; however, there are some spaces for authors to enhance its quality further, which are as follows:

Lines 37-41: There is no need to insert information about the open hydroponics system and its effects on environmental pollution. Therefore, the critical point is to insert the specific details on the updated strategy for increasing and improving the growth and yield Paprika Cultivation in a Closed Coir-Substrate Hydroponic System in South Korea’s Winter Cropping Season. In addition, what are the updated studies about improving the nutrient solution for the production of paprika under a closed hydroponics system? Finally, the authors should explain why they selected this paprika for this study.

1-Lines 72-75: The authors mentioned that producing paprika under closed hydroponic systems makes it challenging to provide proper environmental conditions to grow. Therefore, the authors should determine whether this happened under controlled environmental conditions such as controlled root zone, temperature, relative humidity and other factors.

Also, the authors should insert more information about producing the paprika in coir-Substrate under hydroponic and why they selected a coir substrate.

Therefore, the critical point is to insert the specific details on the updated strategy of development and management of nutrient solution compositions for increasing and improving the growth, yield and nutritional value of paprika, such as mineral concentration, phytochemical and secondary metabolite for Closed Coir-Substrate Hydroponic System in South Korea’s Winter Cropping Season.

Lines 86-90: Please remove and re-write the main aims of this study in one statement.

MATERIALS AND METHODS

Line 92: for the Subtitle Plant materials and experimental conditions, please change it to experimental and site conditions. Please insert a new subtitle for plant materials

The experiment was conducted from august 2014 to July 2017. Therefore, please determine how many seasons or life cycles are for producing paprika.

Please insert new subtitles as follows:

Plant materials

Electrical conductivity and pH

Closed hydroponics system setup

Nutrient solution preparation and management

Experimental design and treatments.

It is difficult to understand your experimental setup and nutrient solution treatments. Therefore, the materials and methods are missing these points.

1.      The schematic diagram of the experimental setup of the experimental layout is not clear.

2.      The number of replication

3.      The life cycle duration of each experiment is not mentioned.

4.      What are the physiochemical properties of coir that were used in this experiment?

5.      I can not understand what the authors did from 2014 until 2017. Therefore, please include the following details:

The experiment started on day month year and ended on day month year. So, for example, The experiment started on 22 June 2019 and ended on 22 September 2019 the first cycle and for the second cycle started on 22 June 2020 and ended on 22 September 2020 etc.

Other comments:

-For Table 1, what do the authors mean by mg/L and Group 1-2, Group 3-6. Please change the composition of the nutrient solution to ppm.

- For Table 2. What do the authors mean by mg/L.

- Why the authors did not control the EC.

- For Table 4, what do the authors mean by kg/10a?

- Please insert the marketable yield for kg/m2 in Table 4.

Please correct the references according to journal format.

Author Response

Response to Reviewer 2 Comments

Thank you for reviewing the manuscript (Development of Nutrient Solution Compositions for Paprika Cultivation in a Closed Coir-Substrate Hydroponic System in South Korea’s Winter Cropping Season) and for your helpful suggestions. We revised the manuscript, and have submitted the revised one. Note that references to lines are with regard to the manuscript as it was received, and these have shifted as changes were made.

<ABSTRACT>

Point 1: (Lines 16-18) Please Re-write the first sentence must be a rationale and please write a problem statement for this study.

Response 1: The first sentence has been revised and added a problem statement (lines 16-20).

Point 2: (Lines 18-21) The objective of the study not clear. Please write in one paragraph the main objectives.

Response 2: The objective of the study has been corrected clearly (Lines 19-21).

Point 3: (Lines 21-23) Please do not use abbreviation in the abstract such as IKC, and PHRS.

Response 3: The abbreviation in the abstract has been explained in brackets (Line 29, 33).

Point 4: The authors should mention the treatments and experimental design to explain the main findings. Please insert the main results for the whole study.

Response 4: The abstract has been rewritten according to your comments. The number of words that can be written to an Abstract is limited. Please let us know if there's anything that needs to be supplemented.

< INTRODUCTION>

Point 5: (Lines 37-41) There is no need to insert information about the open hydroponics system and its effects on environmental pollution. Therefore, the critical point is to insert the specific details on the updated strategy for increasing and improving the growth and yield of Paprika Cultivation in a Closed Coir-Substrate Hydroponic System in South Korea’s Winter Cropping Season. In addition, what are the updated studies about improving the nutrient solution for the production of paprika under a closed hydroponics system? Finally, the authors should explain why they selected this paprika for this study.

Response 5: The sentences (Lines 37-41; original version) related to the open hydroponic system have been deleted. We inserted the latest research on nutrient solution for paprika production under closed hydroponic systems (Lines 53-70). In addition, the reason for paprika's selection in this study was explained (Lines 71-80) and a strategy for increasing paprika yields in a closed coir-substrate hydroponic system in Korea’s winter cropping season was inserted (Lines 104-120).

Point 6: Lines 72-75: The authors mentioned that producing paprika under closed hydroponic systems makes it challenging to provide proper environmental conditions to grow. Therefore, the authors should determine whether this happened under controlled environmental conditions such as controlled root zone, temperature, relative humidity and other factors.

Response 6: The sentences on environmental control and the cropping season for paprika cultivation in Korea have been rewritten to make it easier to understand (Lines 81-90).

Point 7: Also, the authors should insert more information about producing the paprika in coir-substrate under hydroponic and why they selected a coir substrate.

Response 7: Information on the coir-substrate hydroponics and why the coir-substrate was selected for paprika cultivation have been inserted (Lines 81-90).

Point 8: Therefore, the critical point is to insert the specific details on the updated strategy of development and management of nutrient solution compositions for increasing and improving the growth, yield, and nutritional value of paprika, such as mineral concentration, phytochemical and secondary metabolite for Closed Coir-Substrate Hydroponic System in South Korea’s Winter Cropping Season.

Response 8: The strategy of this study for increasing paprika yields in a closed coir-substrate hydroponic system in Korea’s winter cropping season has been inserted (Lines 104-120). Research is currently underway on the management of nutrient solution to increase and improve the nutritional value of plants, such as phytochemicals, vitamins, etc.

Point 9: Please remove and re-write the main aims of this study in one statement.

Response 9: The main purpose of this study has been rewritten in one sentence. (Lines 120-123)

<MATERIALS AND METHODS>

Point 10: Line 92: for the Subtitle Plant materials and experimental conditions, please change it to experimental and site conditions. Please insert a new subtitle for plant materials

Response 10: The subtitle was changed to ‘experimental site and conditions’, and ‘plant materials’ were inserted as a new subtitle (Line 126)

Point 11: The experiment was conducted from august 2014 to July 2017. Therefore, please determine how many seasons or life cycles are for producing paprika.

Response 11: The sentences about the cultivation period and life cycle of each experiment have been inserted (Lines 127-134).

Point 12: Please insert new subtitles as follows: Plant materials, Electrical conductivity and pH, Closed hydroponics system setup, Nutrient solution preparation and management, Experimental design and treatments.

Response 12: Thank you for your suggestion. New subtitles have been inserted with reference to your suggestion, and the contents have been supplemented. Please see section ‘2. Materials and Methods' (p. 3-8)

Point 13:

It is difficult to understand your experimental setup and nutrient solution treatments. Therefore, the materials and methods are missing these points.

  1. The schematic diagram of the experimental setup of the experimental layout is not clear.
  2. The number of replication
  3. The life cycle duration of each experiment is not mentioned.
  4. What are the physiochemical properties of coir that were used in this experiment?

Response 13:

  1. ‘Materials and methods’ have been rewritten according to your suggestion, and figures and explanations in the text have been added.
  2. The number of replications or repetitions has been checked again (Lines 213-215, Lines 241-243, Lines 305-307, Line 295)
  3. The sentences about the cultivation period and life cycle of each experiment have been inserted (Lines 127-134).
  4. The physicochemical properties of the coir-substrate used in this experiment have been presented (Lines 156-159).

Point 14: I cannot understand what the authors did from 2014 until 2017. Therefore, please include the following details: The experiment started on day month year and ended on day month year. So, for example, The experiment started on 22 June 2019 and ended on 22 September 2019 the first cycle and for the second cycle started on 22 June 2020 and ended on 22 September 2020 etc.

Response 14: Thank you for your suggestion. The sentences about the cultivation period and life cycle of each experiment have been inserted (Lines 127-134).

<Other comments>

Point 15: For Table 1, what do the authors mean by mg/L and Group 1-2, Group 3-6. Please change the composition of the nutrient solution to ppm.

Response 15: In Table 1, me/L was changed to ppm, and the meaning of Group has been explained as a fruit set period (Table 1).

Point 16: For Table 2. What do the authors mean by mg/L

Response 16: In Tables 2, me/L was changed to ppm

Point 17: Why the authors did not control the EC

Response 17: In this study, the EC was controlled (Lines 265-271).

Point 18: For Table 4, what do the authors mean by kg/10a?

Response 18: The unit of kg/10a was changed to kg/m2

Point 19: Please insert the marketable yield for kg/m2 in Table 4.

Response 19: The unit of marketable yield has been changed to kg/m2

Point 20: Please correct the references according to the journal format.

Response 20: The references have been corrected according to the journal format

Reviewer 3 Report

The manuscript describes the development of nutrient solution compositions for paprika cultivation in a closed coir-substrate hydroponic system in South Korea’s winter cropping season.

The text is well-written; some minor points are raised below.

1.           Please, insert a photo of a) capsicum annum plant and b) the venlo-type glasshouse.

2.           An extra explanation is needed why new nutrient solutions have been developed given that in the glasshouse the “weather” can be adjusted.

3.           Please, provide the possible economic benefits derived from the application of these new nutrient solutions.

4.           Abstract: NIHHS, PBG, IKC, PHRS and RZ should be explained.

5.           Please, replace me/L with meq/L in all text.

6.           Also, “5–10 % and 10–15 % readjustment” needs further explanation or better rephrazing.

Author Response

Response to Reviewer 3 Comments

Thank you for reviewing the manuscript (Development of Nutrient Solution Compositions for Paprika Cultivation in a Closed Coir-Substrate Hydroponic System in South Korea’s Winter Cropping Season) and for your helpful suggestions. We revised the manuscript, and have submitted the revised one. Note that references to lines are with regard to the manuscript as it was received and these have shifted as changes were made.

Point 1: Please, insert a photo of a) capsicum annum plant and b) the venlo-type glasshouse.

Response 1: Photos of paprika(capsicum annum L.) plants and Venlo-type glasshouse have been inserted in the section of Materials and Methods.

Point 2: An extra explanation is needed why new nutrient solutions have been developed given that in the glasshouse the “weather” can be adjusted

Response 2: The necessity of developing the new nutrient solution compositions has been presented and the sentences on environmental control and the cropping season for paprika cultivation in Korea have been rewritten to make it easier to understand (Lines 71-92, Lines 93-103).

Point 3: Please, provide the possible economic benefits derived from the application of these new nutrient solutions.

Response 3: Possible economic benefits derived from the application of closed hydroponics and new nutritional solutions were presented in the section of ’Conclusions’.

Point 4: (Line 16-33) Abstract: NIHHS, PBG, IKC, PHRS, and RZ should be explained.

Response 4: The abstract has been rewritten according to the reviewers' comments. Therefore, some of the content has been deleted and the abbreviation has been explained in brackets (Lines 29, 33).

Point 5: (Tables 1 and 2) Please, replace me/L with meq/L in all text.

Response 5: In Tables 1 and 2, me/L has been changed to ppm

Point 6: (Lines 188-189) Also, “5–10 % and 10–15 % readjustment” needs further explanation or better rephrazing.

Response 6: The explanation of “5–10 % and 10–15 % readjustment” has been supplemented for a better understanding (Lines 232-240).

Round 2

Reviewer 2 Report

The authors revised the manuscript according to my comments. 

Author Response

We revised the manuscript, and have submitted the revised one. Note that references to lines are with regard to the manuscript as it was received and these have shifted as changes were made.

Point 1: (Lines 391437) Data in table 3 are partly replicated in figures 4 and 5 and therefore, table 3 is partly redundant. The delta ion is derived from the difference between the nutrient concentration in the root zone and the nutrient concentration in the nutrient solution, but this amount is not scientifically relevant. I advise the authors to calculate for each period (t1-t0) the apparent uptake concentration for each nutrient (Cup):

Cup= [Cir x Vup - (Vrz1 x Crz1 -Vrz0 x Crz0)  - Vmt (Cmt1-Cmt0)] / Vup

where; Vrz and Crz= volume and concentration  of nutrient solution retained in the substrate at the time t1 and t0; Vmt and Cmt = volume and concentration of the recirculating nutrient solution present in drainage and mixing tanks and Vup is the crop evapotranspiration
Vup=evapotranspiration volume
So, table 3 must be removed or substituted with the new uptake concentrations calculated.
For more details on the concept of Cup please consul also chapter 13 of the book: Sonneveld and Voogt 2009. Plant Nutrition of Greenhouse Crops.

Response 1: Thank you for your kind suggestion. As you pointed out, the data in Table 3 are partially overlapping in Figures 4 and 5. The delta ion values used in this study differ from the apparent absorption concentration of each nutrient (cup) for each period (t1-t0), a concept commonly used in hydroponic research in Korea, meaning the differences between the ion concentrations in the root zone (or drainage) and with irrigation. Now that the research project has already been completed, and we cannot conduct a new experiment again, we eventually removed Table 3, leaving only some content for discussion of the results (lines 392–407).

Point 2: (Lines 460461, 636638) Furthermore, the statistical analysis of tables 4 and 5 is insufficient. You reported the statistical analyses at two factors, but when the interaction is positive, the means of the interaction must be separated inserting different letters when the means are statistically different. Tables 4 and 5 must report also the average mean of the main factors of ANOVA analysis: when the means are statistically different, a different letter must be inserted near the value (Tables 4 and 5)

Response 2: (Lines 441446, 623-628) As you suggested, if the interaction between the two factors is positive in Table 4 and Table 5, different characters were inserted. We also reported the average mean of the main factors in ANOVA, and if the mean is statistically different, we inserted different characters near the value (Tables 3 and 4).

Point 3: (Line 459, Line 635) please check the captions of tables and figures. The reader must have the necessary information from the captions to understand the figure or the table without reading the text. Please improve the captions of Tables 4 and 5.

Response 3: (Tables 4 and 5) The captions in Tables 4 and 5 have been improved by adding more necessary information needed (Tables 3 and 4).